# Oil Palm Yield Estimation Based on Vegetation and Humidity Indices Generated from Satellite Images and Machine Learning Techniques

Fernando Watson-Hernández [1],*, Natalia Gómez-Calderón [1] and Rouverson Pereira da Silva [2]

1 School of Agricultural Engineering, Instituto Tecnológico de Costa Rica, Cartago 30101, Costa Rica; ngomez@itcr.ac.cr
2 Department of Engineering and Mathematical Sciences, School of Agricultural and Veterinarian Sciences, São Paulo State University (Unesp), Jaboticabal, São Paulo 14884-900, Brazil; rouverson.silva@unesp.br
* Correspondence: fwatson@itcr.ac.cr; Tel.: +50-68-4746643

**Abstract:** Palm oil has become one of the most consumed vegetable oils in the world, and it is a key element in profitable global value chains. In Costa Rica, oil palm cultivation is one of the three crops with the largest occupied agricultural area. The objective of this study was to explain and predict yield in safe time lags for production management by using free-access satellite images. To this end, machine learning methods were performed to a 20-year data set of an oil palm plantation located in the Central Pacific Region of Costa Rica and the corresponding vegetation indices obtained from LANDSAT satellite images. Since the best correlations corresponded to a one-year time lag, the predictive models Random Forest (RF), Least Absolute Shrinkage and Selection Operator (LASSO), Extreme Gradient Boosting (XGBoost), Recursive Partitioning and Regression Trees (RPART), and Neural Network (NN) were built for a Time-lag 1. These models were applied to all genetic material and to the predominant variety (AVROS) separately. While NN showed the best performance for multispecies information ($r^2$ = 0.8139, NSE = 0.8131, RMSE = 0.3437, MAE = 0.2605), RF showed a better fit for AVROS ($r^2$ = 0.8214, NSE = 0.8020, RMSE = 0.3452, MAE = 0.2669). The most relevant vegetation indices (NDMI, MSI) are related to water in the plant. The study also determined that data distribution must be considered for the prediction and evaluation of the oil palm yield in the area under study. The estimation methods of this study provide information on the identification of important variables (NDMI) to characterize palm oil yield. Additionally, it generates a scenario with acceptable uncertainties on the yield forecast one year in advance. This information is of direct interest to the oil palm industry.

**Keywords:** crop yield; google earth engine; neural network; random forest; simulation

## 1. Introduction

Palm oil has become one of the most consumed vegetable oils in the world due to the better extraction performance of its oil as compared to other types of oil-bearing crops such as soybeans, rapeseed or sunflower [1]. Palm oil is also a key element for profitable global value chains [2]. In Costa Rica, oil palm crops are one of the three crops with the largest occupied agricultural area. The forecasting prior to the harvest of the fresh fruit bunch is an important means to evaluate total production, which, in turn, provides useful information for the decision making related to storage, distribution, and marketing budget [3,4].

The use of empirical methods when estimating yield quantity and predicting crop production and loss leads to errors related to human factors [5]. In order to tackle the deficiencies of these methods, appropriate monthly yield forecast by means of artificial intelligence models have been created. These models describe the quantitative relation between meteorological variables with time lags and information related to the fresh fruit bunch, considering the yield of young-mature oil palm for the first six years of harvest [6].

On the other hand, being perennial trees, oil palms have a canopy structure more similar to a forest than to other agricultural crops. Therefore, remote sensing of this crop can be based on aerial or satellite images [4,6,7]. Rodríguez et al. [4] have shown that crop density estimation by sub-pixels is feasible. In agricultural applications in particular, density maps allow a more advanced analysis than that of crop land coverage binary maps. This is important for oil palm plantations, where the distance between trees is known to correlate with production yield.

To overcome the limitations of the monitoring images obtained, multiple spectral bands were used in modeling the colors, estimating vegetation indices (VIs), or in doing a spectral mixture analysis (SMA) [7]. Observational data based on spectral reflectance has been widely used to monitor crop conditions and estimate their yield [8]. The bank of information per pixel and the information from the crops allow the creation of correlations from algorithms that facilitate the analysis of the data. Khanal et al. [9] recommend evaluating various machine learning algorithms to improve the precision of the prediction estimates as well as the evaluation of soil parameters and crops yield. This is done by using data division relations for the training and validation of the model with a 7:3 or 8:2 ratio [10–13]. Amirruddin et al. [13] state that using machine learning techniques to manage data is a promising field for the evaluation of oil palm crops since it reduces costs, time, and intensive work for the wide plantation areas.

Since the LANDSAT satellite was launched into orbit in 1972, free-access multispectral satellite images are available for the region. The LANDSAT images provide information of up to 11 wavelengths (bands), with resolutions of between 15 m and 100 m, which are acceptable ranges due to the fact that the management areas for oil palm cultivation in Costa Rica exceed four hectares [14]. Due to the climatic variability of the Central Pacific region of Costa Rica, it is important to find relationships between images multispectrality, yield, climatic and edaphic conditions, and the age of the crop. This allows explaining and predicting production in safe time lags for the management of production.

The objective of this work was to generate a tool that facilitates the prediction of crop yield. To this end, several statistical models were designed to estimate the annual production of the oil palm crop by means of the vegetation and humidity indices generated from the Landsat 5, 7 and 8 images in the Central Pacific of Costa Rica and the time lag with better correlation between index and crop yield.

## 2. Materials and Methods

The study was carried out with the information of an oil palm crop from an agribusiness located in the Central Pacific of Costa Rica (Figure 1). The crop has a 9 × 9 m field spacing between plants. The analyzed plantation consisted mainly of the AVROS variety (40%), EKONA (21%), and others in the process of being replaced by AVROS (39%). One hundred and three productive units (103 PU) were considered, with information of between 16 and 20 years of cultivation on genetic variety, crop yield (t/ha) and year of sowing. Such variables were averaged monthly for each year and the best correlation was identified for time lags of one, two, three or four years.

For the productive units considered, the images Landsat 5, 7 and 8 Collection 1 Tier 1 calibrated top-of-atmosphere (TOA) reflectance [15] corresponding to a period between January 1996 and December 2016 were obtained, in the ranges described in Table 1. Data management was done through a Java Script code on Google Earth Engine (GEE). The code cyclically generates 12 indices of both vegetation and humidity for each image of the collection (Table 2), allowing to correlate the spectral information of the images with the biophysical properties of the vegetation cover [16]. This generates a new 20 year-history collection with an inter-annual resolution for each of the indices used. As input, the tool requires information of the geometry of each one of the 103 PUs, which constitute the minimum information unit used about production. From the layers of indices and the delimited areas, a statistical description of the behavior of the index is generated in each PU for each date of image capture. The information consists of minimum and maximum

values, quartiles 25th and 75th, means, medians and standard deviations. The information is ordered chronologically in a file for each PU with a total of 84 variables. For the selection of the photographs used, there was no discrimination within the range of dates established by any parameter, so that the models generated were in charge of assigning greater or lesser weights to each variable when using the values of maximum, minimum, percentiles and standard deviation of the vegetation index within the PUs.

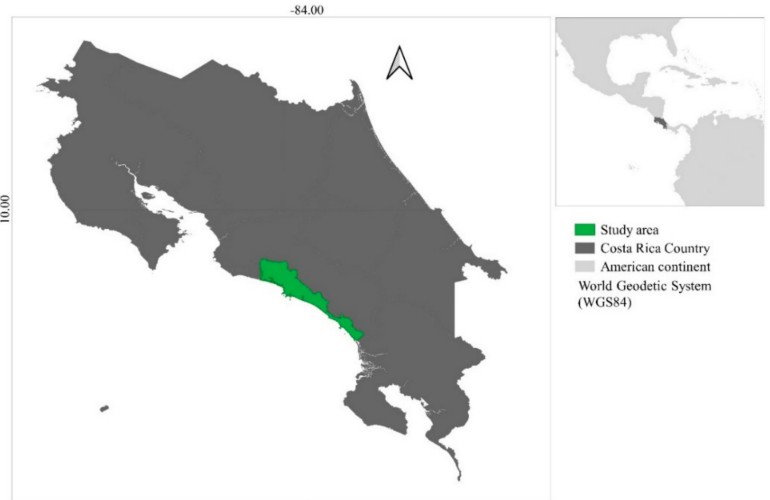

**Figure 1.** Location of the oil palm plantation under study.

**Table 1.** The standard spectral bands of the optical range of the images used in this study.

| Bands | Wavelength (μm) | | |
|---|---|---|---|
| | **Landsat 5** | **Landsat 7** | **Landsat 8** |
| 1 (BLUE) | 0.45–0.52 | 0.441–0.514 | |
| 2 (BLUE) | | | 0.442–0.5120 |
| 2 (GREEN) | 0.52–0.60 | 0.519–0.601 | |
| 3 (GREEN) | | | 0.533–0.590 |
| 3 (RED) | 0.63–0.69 | 0.631–0.692 | |
| 4 (RED) | | | 0.636–0.673 |
| 4 (NIR) | 0.76–0.90 | 0.772–0.898 | |
| 5 (NIR) | | | 0.851–0.879 |
| 5 (SWIR) | 1.55–1.75 | 1.547–1.749 | |
| 6 (SWIR) | | | 1.566–1.651 |

RStudio (2018) was used to merge information about the crops yield with the vegetation and humidity indices. The process consisted of changing the time scale from monthly data to annual averages, eliminating the missing data, and overlapping the data of indices with time lags of one, two, three and four years (time lags = 1, 2, 3 and 4 years) with respect to the series of performance data.

In selecting the time lag to work with, the individual correlation was determined for each index considering the performance of each of the proposed time lags. The time lag with the best correlation was used to build the AI models. The models used were Random Forest (RF), XGBoost algorithm (XGBoost), LASSO regression (LASSO), Recursive Partitioning and Regression Trees (RPART), and Neural Network (NN); the packages used were randomForest, xgb.train, glmnet, rpart and neuralnet, respectively, to which subroutines were generated in RStudio. The configuration settings of the proposed models are in Table 3.

**Table 2.** Multispectral indices from spectral channels using the Landsat 5, Landsat 7 and Landsat 8 collections.

| Variable | Index | Equation | Source |
|---|---|---|---|
| Vegetation | ARVI | $(NIR - (2 \cdot RED) + BLUE)/(NIR + (2 \cdot RED) + BLUE)$ | [17] |
| Vegetation | AVI | $[NIR \cdot (1 - RED) \cdot (NIR - RED)]^{1/3}$ | [18] |
| Vegetation | EVI | $2.5 \cdot (NIR - RED)/(NIR + 6 \cdot RED - 7.5 \cdot BLUE + 1)$ | [19] |
| Vegetation | GCI | $(NIR/GREEN) - 1$ | [17] |
| Vegetation | GNDVI | $(NIR - GREEN)/(NIR + GREEN)$ | [19] |
| Vegetation | NDVI | $(NIR - RED)/(NIR + RED)$ | [19] |
| Vegetation | NPCRI | $(RED - BLUE)/(RED + BLUE)$ | [18] |
| Vegetation | SAVI | $1.5 \cdot (NIR - RED)/(NIR + RED + 0.5)$ | [19] |
| Vegetation | SIPI | $(NIR - BLUE)/(NIR + BLUE)$ | [19] |
| Water | MSI | $(SWIR/NIR)$ | [20] |
| Water | NDMI | $(NIR - SWIR)/(NIR + SWIR)$ | [20] |
| Water | NDWI | $(GREEN - NIR)/(GREEN + NIR)$ | [20] |

**Table 3.** Parameters used in the configuration of the proposed models.

| Model | Parameters | Description | Source |
|---|---|---|---|
| LASSO | alpha | The elasticnet mixing parameter, with $0 \leq \alpha \leq 1$ | [21] |
| | lambda | Regularization hyperparameter | |
| RF | ntree | Number of trees to grow | [22] |
| | mtry | Number of variables randomly sampled as candidates at each split | |
| XGBoost | max.depth | Maximum depth of a tree | [23] |
| | nrounds | The number of decision trees in the final model | |
| | nthread | Number of parallel threads used to run XGBoost | |
| | objective | Specify the learning task and the corresponding learning objective | |
| RPART | minsplit | The minimum number of observations that must exist in a node in order for a split to be attempted. | [24] |
| | minbucket | The minimum number of observations in any terminal node | |
| | cp | Complexity parameter | |
| NN | threshold | A numeric value specifying the threshold for the partial derivatives of the error function as stopping criteria | [25] |
| | stepmax | The maximum steps for the training of the neural network | |
| | algorithm | A string containing the algorithm type to calculate the neural network | |

As seen in Figure 2, the dependent variable was the performance with a time lag of one year and the predictor variables were the vegetation and humidity indices. The set was then randomly divided into the training (Data Training) and validation (Data Test) stages in a 7:3 ratio. The Data Test was further subdivided into two series of information. The first consisted of data from all varieties of oil palm (genetic material) produced in the time interval under study, however, the second consisted of the information from the AVROS palm variety, given that it is the predominant genetic material.

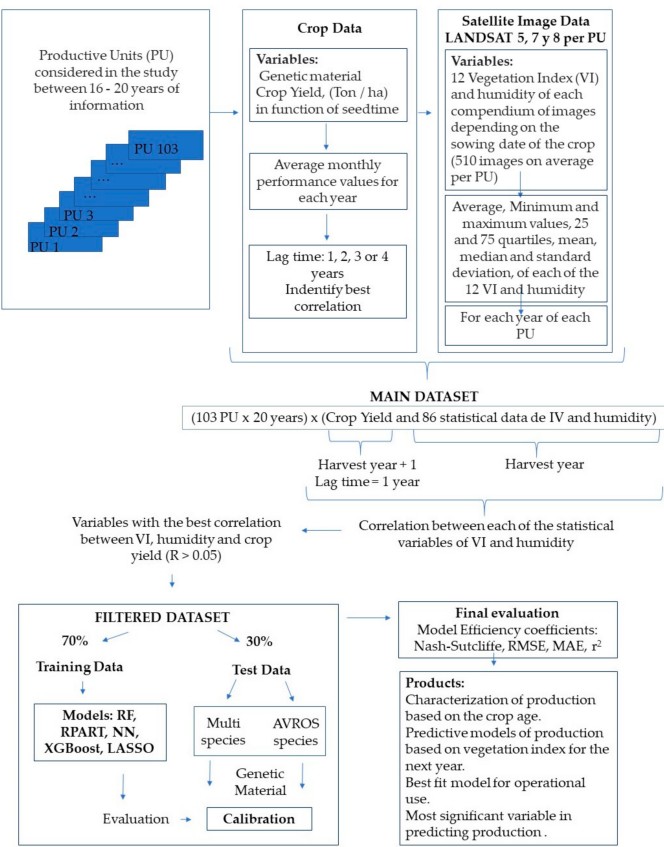

**Figure 2.** Methodological diagram of the analysis of the information and model's sensitivity.

The evaluation of the models was carried out by applying the efficiency coefficient of the Nash–Sutcliffe model (NSE) (Equation (1)), the root mean square error (RMSE) (Equation (2)), the mean absolute error coefficient (MAE) (Equation (3)), and the coefficient of determination ($r^2$) (Equation (4)).

$$\text{NSE} = 1 - \frac{\sum_{i=1}^{n}\left(y_i^{obs} - y_i^{sim}\right)^2}{\sum_{i=1}^{n}\left(y_i^{obs} - \overline{y}^{obs}\right)^2} \tag{1}$$

$$\text{RMSE} = \sqrt{\frac{1}{n}\sum_{i=1}^{n}\left(y_i^{obs} - y_i^{sim}\right)^2} \tag{2}$$

$$\text{MAE} = \frac{1}{n}\sum_{i=1}^{n}\left|y_i^{obs} - y_i^{sim}\right| \tag{3}$$

$$r^2 = \left(\frac{\sum\left[\left(y_i^{obs} - \overline{y}^{obs}\right)\left(y_i^{sim} - \overline{y}^{sim}\right)\right]}{\sqrt{\sum\left(y_i^{obs} - \overline{y}^{obs}\right)^2 * \sum\left(y_i^{sim} - \overline{y}^{sim}\right)^2}}\right)^2 \tag{4}$$

where $n$ is the amount of data available for the study, $y_i^{obs}$ corresponds to the information on the crop yield obtained for the interval under study, $\overline{y}^{obs}$ is the average of the crop yield obtained for the interval under study, $y_i^{sim}$ is the simulated crop yield, and $\overline{y}^{sim}$ is the average simulated crop yield.

The best model for each of the data series used in the validation process (AVROS and Multi Species) was determined by comparing the values of the efficiency and error coefficients obtained for each model (RF, LASSO, XGBoost, RPART, NN). For the best

qualified model according to the data series of the data test, the importance of the vegetation and humidity indices was determined, following the procedure described below.

### 2.1. Variable Importance Random Forest Model

In determining the importance of the variables for the Random Forest model, the Mean Decrease Gini ($I_G(\theta)$) method was used, as shown in Equation (5). The Mean Decrease Gini is a measure that dimensions the importance of the variables based on the Gini impurity index ($i(\tau)$) used to calculate the divisions of the trees. A loss function (mse) is used where, by comparison, it is established that the most useful variables achieve greater increases in the purity of the nodes [26].

$$I_G(\theta) = \sum_T \sum_\tau \Delta i_\theta(\tau, T) \qquad (5)$$

where, considering all the variables ($\theta$) used in each node ($\tau$) within the trees ($T$) of the Random Forest model, the number of optimal divisions is established, from which the decrease in the Gini impurity $\Delta i_\theta(\tau, T)$ is calculated, which is registered and accumulated for each node $\tau$ of each tree ($T$) of the Random Forest model individually for each of the variables ($\theta$).

### 2.2. Variable Importance Neural Network Model

To determine the importance of the variables of the Neural Network model, Garson's algorithm method [27] was used. This identifies the relative importance of the independent variables in a NN by deconstructing the weights assigned by the model, according to the Equation (6). The relative importance of an independent variable in response to the dependent variable is identified by locating all connections through the weighting of the nodes of interest. This procedure is repeated for all the independent variables used in the model [28].

$$CR_{ik} = \frac{\sum_{j=1}^{L} \left( \frac{w_{ij}}{\sum_{r=1}^{N} w_{rj}} v_{jk} \right)}{\sum_{i=1}^{N} \left( \sum_{j=1}^{L} \left( \frac{w_{ij}}{\sum_{r=1}^{N} w_{rj}} v_{jk} \right) \right)} \qquad (6)$$

where, $CR_{ik}$ represents the percentage of influence of each input independent variable $i$ on the output dependent variable $\sum_{r=1}^{N} w_{rj}$, being the sum of the weights that connect the input layer $i$ and the neuron $j$; $N$ corresponds to the total of input variables; $L$ corresponds to the total of the hidden layer; $v_{jk}$ that corresponds to the weights of the connection between the input neuron $j$ and the input vector $k$.

The information on the yield of the oil palm fruit production used has a normal behavior (Figure 3).

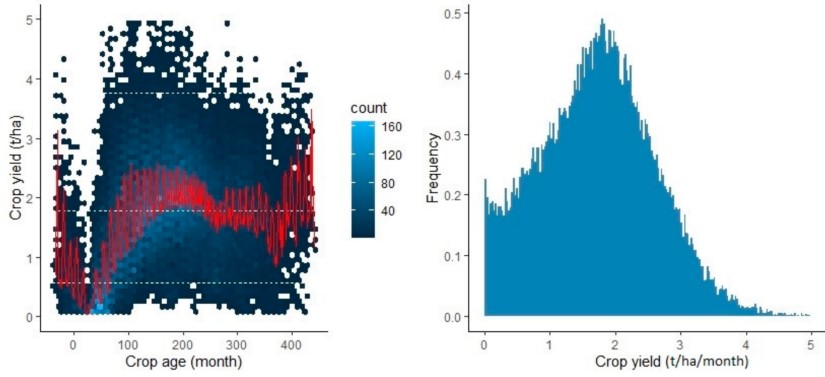

**Figure 3.** Characterization of the production behavior of oil palm.

## 3. Results

The analysis of the yield of the 103 PUs used in the creation of the models determined that, between month 0 and 84, there is a growth in yield of 0.26 t/ha per year. Thereafter, a constant production is relatively maintained up to approximately year 20 of the crop, where the most frequent production value is 1.92 t/ha per month (Figure 3).

### 3.1. Correlations between Vegetation and Humidity Indices for Different Time Lags

Figure 4 shows the value of the correlation coefficient for Time-lags of one, two, three and four years. The 15 vegetation indices with the highest absolute correlations with crop yield were selected. Positive values indicate that the higher the index value, the better the plantation yield and, on the contrary, indices that show water stress (MSI) [29] or the identification of senescence in the plantation (NPCRI) [30] will have a inversely proportional relationship.

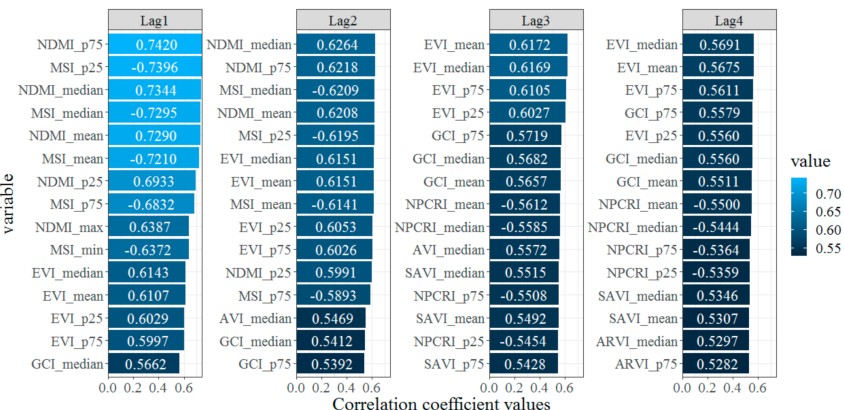

**Figure 4.** Crop yield correlation coefficients with the variables generated for time lags of one, two, three and four years.

Moisture-related indices (NDMI, MSI), either by amount of water or by plant stress, show the highest correlations with yield for lags of one and two years (Lag-time 1 and Lag-time 2). After the three-year delay time (Time Lag-3), humidity is not significant in the analysis, and the vegetation indices relative to chlorophyllous activity (EVI, GCI, NPCRI) show the best yield relations.

The correlation values of the indices relative to chlorophyllous activity does not show variation in any of the delay times studied (Figure 4). In any case, the highest value reached is *r* = 0.63 for a lag of three years.

The best correlations were obtained for the Time-Lag of one year, as its first six variables present values above *r* = 0.70, while the lags from two to four years do not exceed *r* = 0.63. Therefore, the prediction models were built for a Time-lag 1.

### 3.2. Coefficients of Crop Production Prediction Models

Machine learning models were built according to the RF, LASSO, XGBoost, RPART and NN structures. Table 4 shows the results obtained in the calibration of the models. For the Values *NSE* and *r²*, acceptability ranges according to [31] were followed; whereas for RMSE and MAE, the average production yield (1.92 t/ha) was used. A 10% error was considered acceptable, and more than 40% unsatisfactory. For the RF model, three different values according to the number of trees (ntree) were tested: 5000, 15,000 and 10,000. The best performance according to the statistical evaluation coefficients used was obtained for ntree = 10,000.

**Table 4.** Calibration evaluation coefficients for the models used.

| Model | Variable/Value | | NSE | RMSE | MAE |
|---|---|---|---|---|---|
| LASSO | lambda = 0.0037 | | 0.7701 | 0.3487 | 0.2679 |
| Neural Network (NN) | stepmax = $1 \times 10^6$ | | Algorithm did not converge | | |
| | stepmax = $1 \times 10^6$ | | Algorithm did not converge | | |
| | stepmax = $1 \times 10^7$ | | 0.7947 | 0.3925 | 0.2514 |
| Random Forest (RF) | ntree = 5000 | | 0.9519 | 0.1668 | 0.1282 |
| | ntree = 10,000 | | 0.9527 | 0.1655 | 0.1266 |
| | ntree = 15,000 | | 0.9518 | 0.167 | 0.1279 |
| Recursive Partitioning and Regression Trees (RPART) | Minsplit = | 5 | 0.6652 | 0.4207 | 0.3217 |
| | | 10 | 0.6652 | 0.4207 | 0.3218 |
| | | 50 | 0.6652 | 0.4207 | 0.3218 |
| | | 100 | 0.6652 | 0.4207 | 0.3217 |
| | | 500 | 0.5495 | 0.488 | 0.3765 |
| | | 1000 | 0.4133 | 0.557 | 0.4364 |
| | Complexity parameter (cp) = minsplit = 10 | 0.01 | 0.6652 | 0.4207 | 0.3218 |
| | | 0.001 | 0.8875 | 0.2438 | 0.1876 |
| | | 0.0001 | 0.9322 | 0.1894 | 0.1324 |
| XGBoost | max.depth = 1 | | 0.6777 | 0.4127 | 0.3180 |
| | max.depth = 2 | | 0.8230 | 0.3059 | 0.2322 |
| | max.depth = 3 | | 0.9034 | 0.2259 | 0.1727 |
| | max.depth = 4 | | 0.9567 | 0.1512 | 0.1147 |
| | max.depth = 5 | | over fitting | | |

The LASSO linear regression method (glmnet package) was included in the analysis as a reference method. The lambda that generated the lowest MSE was used to calibrate it ($\lambda$ = 0.0037).

For RPART, the minsplit parameter was used first to improve performance. The minsplit values 5, 10, 50, 100, 500 and 1000 were tested, and Minsplit = 10 was selected. The model fit continued with the variable complexity parameter (cp) for values of 0.01, 0.001 and 0.0001, being the latter the one used. In the NN model, a stepmax of $1 \times 10^7$ was used as it presented convergence problems for lower values (Figure 4).

*3.3. Calibration (Training) and Validation (Test) of Prediction Models for Crop Production*

As seen in Figure 5, the five models generated have a satisfactory performance for the training, with values of $r^2 \geq 0.7900$, NSE $\geq 0.7900$ and RMSE $\leq 0.3300$ t/ha. The best model in the training was XGBoost with $r^2 = 0.9597$, NSE = 0.9567 and RMSE = 0.1513 t/ha, followed by RF with a slightly lower performance. RPART presented atypical residue values higher than XGBoost and RF (Figure 6), which generates an increase in RMSE = 0.1894 t/ha. Nevertheless, these atypical data in the residues are present in yields between 1–2 t/ha, where the model tends to overestimate palm production in some dispersed points. Its performance, however, improves for values higher than 2 t/ha. Since the average production of the time interval of the crop under study is 1.92 t/ha, the RPART model can be used with a 98% confidence for estimated values greater than the average. The LASSO and NN models presented a variation in the residues greater than RF, XGBoost and RPART, which generated an RMSE higher than 0.3200 t/ha, 59% more than RF, XGBoost and RPART. The LASSO reference model obtained acceptable coefficients of NSE = 0.7701, $r^2 = 0.7702$ during the training process, however the performance was slightly lower than the average of the other machine learning methods. The LASSO model generated coefficient values of RMSE = 0.3487 t/ha, MAE = 0.2679 t/ha that are within the range considered acceptable, yet these errors are higher than the average errors of the best performing machine learning models (RF, XGBoost).

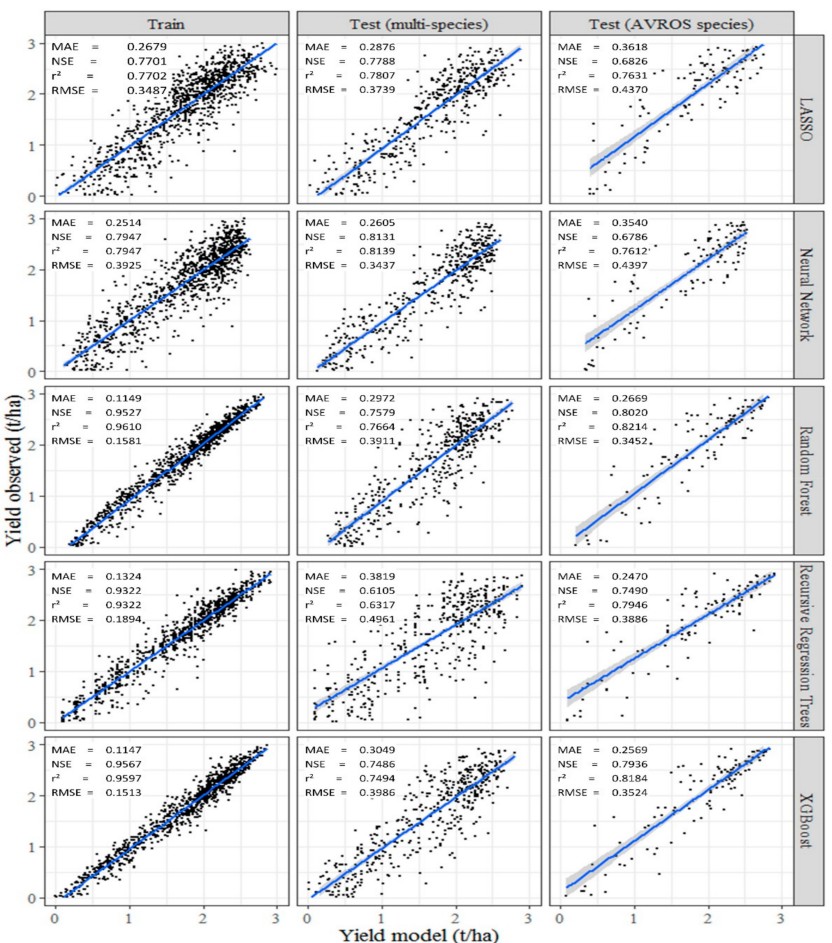

**Figure 5.** Comparison of observed performance with simulated performance in the calibration and validation processes.

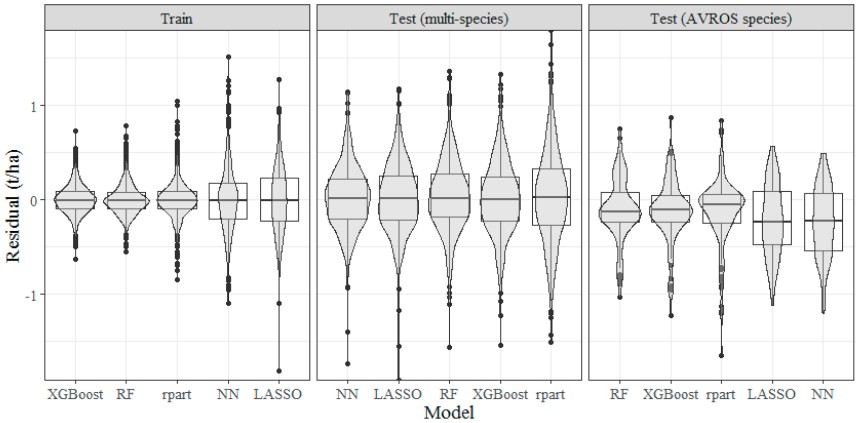

**Figure 6.** Residual distribution of the models in the calibration and validation processes. The dots indicate the value of the RMSE.

For validation the data set consisted of information from multiple palm species including all the genetic materials used in the cultivation between 1996 and 2016 (Multi-species Test). In this stage, the evaluation coefficients of the models decreased, presenting values of $r^2$ between 0.6400–0.8300, NSE between 0.6100–0.8100, and the RMSE between 0.3400–0.5000 t/ha. In descending order of the evaluation coefficients, the best model was NN, followed by LASSO, RF, XGBoost and RPART. RPART presented a higher error than

(RMSE = 0.4961 t/ha) the other models, thus, it is the least recommended model to estimate yield from information that considers multiple species (genetic material).

For the validation based on the information of the crop yield of the AVROS species (24.9% of the total data), the models behaved similarly to the multi-species test, presenting $r^2$ values between 0.7600–0.8100, NSE between 0.6800–0.8000 and RMSE between 0.3400–0.4400 t/ha. However, following the same ordering criterion of the calibration models, the best model was RF, followed by XGBoost, RPART, LASSO and NN. Consequently, NN is the least recommended model when using only the information of the predominant genetic material (AVROS). The selection of the model is influenced by the variability of the genetic material used in planting, as this alters the performance of the machine learning techniques.

### 3.4. Importance of Vegetation Indices Variables in Crop Prediction

Based on the evaluation coefficients of the models in both training and validation together with residue behavior, it is determined that the NN model has the best performance for the multispecies Test, while for the AVROS species test, the RF model shows the best fit. When performing the analysis of the variables of the most important vegetation indices (VIs) to predict from these two models (Figure 7), the median of the NDMI (NDMI_median) stands out as the most important variable for both models.

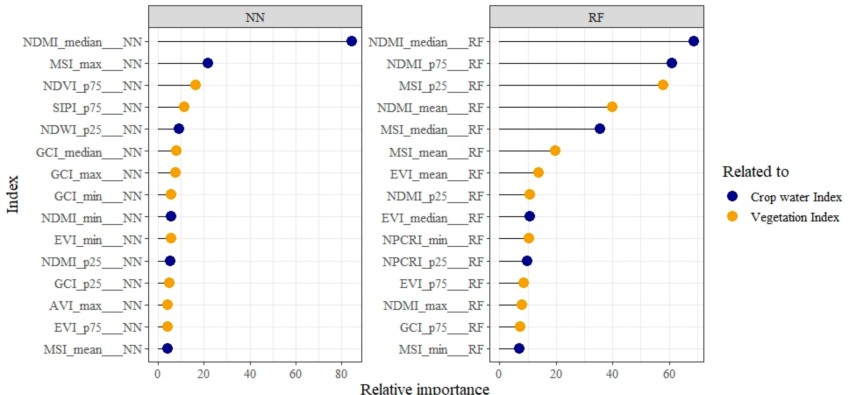

**Figure 7.** Coefficients of the relative importance of the variables used in the two best models.

### 4. Discussion

The most important VIs obtained from the Landsat 5, 7 and 8 satellite images are related to water in the plant. The 75th and 25th percentiles of these VIs improve the performance of the models compared to other studies where they were not considered (Table 5). In the two best models NN (multispecies Test) y RF (AVROS species test) of the fifteen most important variables, six correspond to the percentiles of the VIs (25th and 75th quartiles); that is, the distribution of the available information on the VIs of each PU impacts the performance of the models significantly and they should be considered together with the mean values of the VIs in predicting and evaluating oil palm production.

**Table 5.** Previous research works on oil palm crop yield prediction.

| Method | Variables | Coefficient | Source |
|---|---|---|---|
| Artificial Neural Network | NDVI | $r^2$ = 0.5100 | [32] |
| Genetic algorithm | Historical yield data, cropland information, climatic information, air pollutants | $r^2$ = 0.9400 RMSE = 0.1500 t/ha | [33] |
| Artificial Neural Network | Climatic information | MAE = 0.5300 t/ha RMSE = 0.6800 t/ha | [34] |
| Long short-term memory (LSTM) | Historical yield data | MAPE = 2.7100% | [35] |

The study by Hilal et al. [33] obtained a better performance than in this study (the difference in RMSE is 0.1950 t/ha); however, the data of the model proposed in this study were obtained from satellite images and did not require additional data measured on site, which reduces its difficulty of application.

In the prediction of the performance that considered the totality of the genetic material, the distribution of the residues in the validation stage shows an average value near zero (mean = 0.0300 t/ha) and a homogeneous distribution with a higher concentration between quartiles 25th and 75th, for the NN, RF and XGBoost models. On the other hand, when considering only the variety AVROS, the mean of the residues during validation is less than zero in all the models. Thus, the distribution of the residues for AVROS species test losses homogeneity compared to the Multi Specie Test, which will cause the crop yield to be underestimated in the prediction. Due to this, the NN model for the Multi-specie Test would not differentiate the significance of some variables.

In predicting the production behavior of the oil palm crop with any genetic material by means of machine learning techniques (RF, RPART, LASSO, XGBoost and NN) and vegetation indices variables obtained from Landsat 6, 7 and 8 satellite images, the appropriate time lag is 1 year. The NN model is the one with the best performance for crop information considering multiple genetic planting materials. However, NN is the least recommended model when using only the predominant genetic material information. When considering the predominant genetic material (AVROS), the RF model shows the best performance. On the other hand, RPART presented a higher error than (RMSE = 0.4961 t/ha) the other models, and thus, it is the least recommended to estimate yield from information that considers multiple species.

Our results show that the variability of the genetic material used in planting influences the selection of the model. This is due to performances of machine learning techniques differing based on genetic variability. The vegetation indices related to water in the plant [36] and the data, corresponding to the average and the 25th and 75th percentiles, are the variables that have the greatest influence on the performance of the proposed prediction models. In the best models (NN and RF), six of the fifteen most important variables correspond to percentile vegetation indices (25th and 75th quartiles). This shows that contemplating the distribution of the data within the PUs significantly improves the performance of the models. As a result of this, they must be considered in the prediction and evaluation of oil palm production in the study area.

## 5. Conclusions

The strongest correlations between oil palm yield and VIs were obtained for a lag period of one year. The machine learning methods used to estimate oil palm crop yields as a function of VIs one year in advance showed satisfactory performance. The RF model was the best qualified for predicting oil palm cultivation of the AVROS species (MAE = 0.2669 t/ha, RMSE = 0.3452 t/ha, NSE = 0.8020, $r^2$ = 0.8214), while the NN model was the best when the plantation has multiple species (MAE = 0.2605, RMSE = 0.3437, NSE = 0.8131, $r^2$ = 0.8139). The Normalized Difference Moisture Index (NDMI) is the most relevant variable in the prediction of oil palm cultivation among a total of 12 VIs used, regardless of the type of species under study [37]. The estimation methods of this study can provide information on the identification variables (NDMI) to characterize palm oil yield. Furthermore, it generates a scenario with acceptable uncertainties on the yield forecast one year in advance, which is of direct interest to the palm oil industry.

**Author Contributions:** Conceptualization, F.W.-H. and N.G.-C.; methodology, F.W.-H.; software, F.W.-H.; validation, F.W.-H. and N.G.-C.; formal analysis, F.W.-H., N.G.-C. and R.P.d.S.; investigation, F.W.-H. and N.G.-C.; resources, F.W.-H.; data curation, F.W.-H.; writing—original draft preparation, F.W.-H. and N.G.-C.; writing—review and editing, F.W.-H., N.G.-C. and R.P.d.S.; visualization, N.G.-C.; supervision, F.W.-H. and N.G.-C.; project administration, N.G.-C.; funding acquisition, F.W.-H. All authors have read and agreed to the published version of the manuscript.

**Funding:** This research received no external funding.

**Institutional Review Board Statement:** Not applicable.

**Informed Consent Statement:** Not applicable.

**Acknowledgments:** We thank the oil palm producers of the Central Pacific of Costa Rica for providing the historical records of production yields and the Vice-Rector's Office for Research and Extension of the Technological Institute of Costa Rica for supporting the study.

**Conflicts of Interest:** The authors declare no conflict of interest.

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
