# Peer review of "Oil Palm Yield Estimation Based on Vegetation and Humidity Indices Generated from Satellite Images and Machine Learning Techniques"

_agriengineering, doi:10.3390/agriengineering4010019_

Round 1

Reviewer 1 Report

This paper proposes to use machine learning techniques to build predictive models for palm oil yield in Costa Rica, where the independent variables are obtained from satellite images. By comparing the model-evaluation scores of different models, it turns out that neural network (NN) model performs the best when estimating multiple palm species while random forecast (RF) performs the best when estimating the major palm species. Several variables that have a significant impact on oil yield are identified based on the NN and RF models.

  1. “machine learning correlations” in the abstract may be changed to “machine learning methods.”
  2. The linear regression method (possibly with regularization like LASSO) is directly applicable to predicting the palm oil yield, which can serve as a baseline method.

  3. The machine learning methods considered in this paper are a bit standard, and I am wondering whether some more recently developed methods can produce better fitting results (e.g., Gaussian-process based regression, XGBoost, deep learning method such as long short-term memory model etc).

  4. Regarding the independent variables, since spatial coordinates of the dat are available, which may reflect some unconsidered spatial heterogeneity, I am wondering whether including the spatial coordinates can improve the estimation results. Besides, since the palm species also affect the corp yield, it may also serve as an independent variable.
  5. How about the forecasting performance of these machine learning methods? Certainly the estimation methods in this paper can provide insights on identifying important variables for characterizing the palm oil yield, the one-year ahead forecasting of the yield is of more direct interest to the palm oil industry.  

Author Response

Dear Reviewer

We thank you for your comments. Your observations were considered and integrated into the study. Below, we present the answers to your contributions in the same order as you did.

  1. “Machine learning correlations” in the abstract may be changed to “machine learning methods.”

“Machine learning correlations” was replaced by “Machine learning methods” in the Abstract.

  1. The linear regression method (possibly with regularization like LASSO) is directly applicable to predicting the palm oil yield, which can serve as a baseline method.

The LASSO linear regression method (glmnet package) was included in the analysis.  The model was calibrated to determine the lambda (λ=0.0037) that generated the lowest MSE. 

The LASSO reference model obtained coefficients of NSE=0.7701, r²=0.7702 which are acceptable, but its performance is a little lower than the average of the machine learning models. Similarly, the LASSO model generated errors of RMSE=0.3487 t/ha, MAE=0.2679 t/ha below the threshold considered acceptable but higher than the average of the machine learning models.

  1. The machine learning methods considered in this paper are a bit standard, and I am wondering whether some more recently developed methods can produce better fitting results (e.g., Gaussian-process based regression, XGBoost, deep learning method such as long short-term memory model etc).

The SVM method was eliminated and replaced by the XGBoost method. This method was used because it is the algorithm with which the group of researchers is most familiar. Allowing the inclusion of this in the original algorithm within the 10-day period granted by the journal to present the reviewer's suggestion.

  1. Regarding the independent variables, since spatial coordinates of the data are available, which may reflect some unconsidered spatial heterogeneity, I am wondering whether including the spatial coordinates can improve the estimation results. Besides, since the palm species also affect the crop yield, it may also serve as an independent variable.

Correct, the inclusion of the variables geographical coordinates and genetic material could generate a better fit of the models. However, the interest of the study was always focused on yield prediction and on the definition of which vegetation indices are most relevant for oil palm production. The objective was to be able to monitor the plantation based on vegetation indices, allowing the implementation of corrective measures from one year before harvest.

  1. How about the forecasting performance of these machine learning methods? Certainly, the estimation methods in this paper can provide insights on identifying important variables for characterizing the palm oil yield, the one-year ahead forecasting of the yield is of more direct interest to the palm oil industry.

Correct, in fact the prediction model is being used by some companies. A section of conclusions is included to affirm the prediction capabilities and the indexes that must be monitored to maintain or raise oil palm crop yields.

The content of the article was re-check, and the English grammar was corrected. The professor Dr. Luis Diego Guillen, is a specialist in English teaching at the School of Language Sciences, who9 realized the corrections and improves of the article. For more references you can write to lguillen@itcr.ac.cr

Reviewer 2 Report

The article provides a detailed analysis on different predictive models to showcase a real case study in the Central Pacific Region of Costa Rica. The proposed predictive models Random Forest (RF), Support Vector Machine (SVM), Recursive Partitioning and Regression Trees (RPART), and Neural Network (NN) are well implemented and described in research article. However, following minor changes will make the paper more presentable:

  • In the abstract, the main importance to carry out the research work is not discussed adequately. It needs to be re-summarized. Little clarity is expected from the authors.
  • The major contribution of the proposed work is discussed in the last part of the section 2. The author should mention the work done or included in the article as a list of point. Correct it.
  • Some grammar errors found throughout the article. The authors should re-check the contents and correct the English grammars.
  • The importance of the proposed work should be included inside the Introduction section of the article.
  • The difference between present work and previous works should be highlighted. The author may include a tabular analysis of recent research works in the article. This may include the quality of the paper. The comparative analysis of these techniques must be included in the subsequent section of the article.
  • The caption or label of the table 1 is not appropriate. This should be re-written in a better way to justify the clear meaning of the sentence.
  • The table 1 shows Table 1. Wavelength of the bands and images used. What is the unit of the wavelength band considered in this research work? Mention the appropriate unit for the wavelength.
  • The line number 112 shows the tool “RStudio (2018)” used in the proposed research work. What packages are used to do the image processing and analysis? This information related to additional packages should be highlighted in the article. This will help the readers to explore more in the particular area of work.
  • The configuration settings of the proposed models such as Random Forest (RF), Support Vector Machine (SVM), Recursive Partitioning and Regression Trees (RPART) and Neural Network (NN) must be mentioned in the article. A tabular list will be enough to justify the proposed work.
  • The Table 2 shows a list of Multispectral indices from spectral channels using the Landsat 5, Landsat 7 and Landsat 8 collections. The table 2 is not cited inside the text anywhere in the article. The author should clearly explain the importance of the list mentioned in the table 2. However, no such information is cited inside the article.
  • In the line 270, How the Importance of vegetation indices variables in crop prediction is evaluated? The author must clearly describe in the article.
  • The last section of the article “4. Discussion” must be re-written with adequate postulations. The author must re-write the content with sufficient comparative justifications.
  • The references are too old. Therefore, the authors must include some relevant works from the last three years to justify the importance of the proposed research work. Please, recheck the reference citations. All references must be cited inside the article.
  • Some page numbers and volume numbers in the references (5, 6, 10, 11, 13, 14, 16, 17, 19, 20) are missing. Add the correct information in those references.

Author Response

Dear Reviewer

We thank you for your comments. Your observations were considered and integrated into the study. Below, we present the answers to your contributions in the same order as you did.

  1. In the abstract, the main importance to carry out the research work is not discussed adequately. It needs to be re-summarized. Little clarity is expected from the authors.

The main importance of the research work was described in the Abstract.

  1. The major contribution of the proposed work is discussed in the last part of the section 2. The author should mention the work done or included in the article as a list of point. Correct it.

The list of points requested with the main contributions of the research work were placed in the Conclusions section.

  1. Some grammar errors found throughout the article. The authors should re-check the contents and correct the English grammars.

The content of the article was re-check, and the English grammar was corrected. The professor Dr. Luis Diego Guillen, is a specialist in English teaching at the School of Language Sciences, who9 realized the corrections and improves of the article. For more references you can write to lguillen@itcr.ac.cr

  1. The difference between present work and previous works should be highlighted. The author may include a tabular analysis of recent research works in the article. This may include the quality of the paper. The comparative analysis of these techniques must be included in the subsequent section of the article.

A tabular analysis of recent research works on oil palm yield prediction was included in the Discussion section of the article (Table 5).

  1. The caption or label of the table 1 is not appropriate. This should be re-written in a better way to justify the clear meaning of the sentence.

The label in Table 1 was modified to refer to the range of spectral resolution of the bands and images, which is used to measure the wavelength of the electromagnetic spectrum.

  1. The table 1 shows Table 1. Wavelength of the bands and images used. What is the unit of the wavelength band considered in this research work? Mention the appropriate unit for the wavelength.

The unit of the wavelength of the bands considered in this research work, which corresponds to micrometers (mm), was added in Table 1.

  1. The line number 112 shows the tool “RStudio (2018)” used in the proposed research work. What packages are used to do the image processing and analysis? This information related to additional packages should be highlighted in the article. This will help the readers to explore more in the particular area of work.

The RStudio (2018) packages that were used to do the image processing and analysis were added in the Materials and Methods section of the article.

  1. The configuration settings of the proposed models such as Random Forest (RF), Support Vector Machine (SVM), Recursive Partitioning and Regression Trees (RPART) and Neural Network (NN) must be mentioned in the article. A tabular list will be enough to justify the proposed work.

The parameter settings for each of the models proposed in the study were included in a tabular list (Table 3) in the Materials and Methods section of the article.

  1. The Table 2 shows a list of Multispectral indices from spectral channels using the Landsat 5, Landsat 7 and Landsat 8 collections. The table 2 is not cited inside the text anywhere in the article. The author should clearly explain the importance of the list mentioned in the table 2. However, no such information is cited inside the article.

Table 2, which was missing in the text of the article, was mentioned, and the importance of the list of "Multispectral indices" in Table 2 was also explained.

  1. In the line 270, How the Importance of vegetation indices variables in crop prediction is evaluated? The author must clearly describe in the article.

How the importance of the vegetation and humidity indices was evaluated is described from line 171 to 204.

The results of the importance of these indices in crop prediction are presented from line 267 to 341.

  1. The last section of the article “4. Discussion” must be re-written with adequate postulations. The author must re-write the content with sufficient comparative justifications.

Postulations were added to the Discussion section along with the tabular analysis of recent research works, rewriting a section of the content through comparative justifications.

  1. The references are too old. Therefore, the authors must include some relevant works from the last three years to justify the importance of the proposed research work. Please, recheck the reference citations. All references must be cited inside the article.

  • References that were very old were changed for more recent works (2019 onwards).
  • References [7], [14], [21], [22], [23], [25], [26] and [34], from 2009-2016, were maintained since they are research works that present important information or methodologies used in this article.

  1. Some page numbers and volume numbers in the references (5, 6, 10, 11, 13, 14, 16, 17, 19, 20) are missing. Add the correct information in those references.
  • References [5] and [11] are article in press, these are articles that have been accepted for publication but have not yet been assigned to a publication volume/issue, but can be cited using the DOI. These types of references have been observed in other MDPI journal articles, referencing them according to the American Chemical Society from which the "Reference List and Citations Style Guide for MDPI Journals" is based.
  • References [6], [10], [13], [16], [17], [19], [20], [24], [26], [27], [28], [29], [30], [31] and [37] are articles published in online-only Publishers journals, which have started to use article numbers instead of page numbers. These were referenced according to the instructions of "Reference List and Citations Style Guide for MDPI Journals".
  • Reference [14] is a Book; it was referenced according to the instructions of "Reference List and Citations Style Guide for MDPI Journals".
  • Reference [18] is a Website, it was referenced according to the instructions in "Reference List and Citations Style Guide for MDPI Journals".
  • References [22], [23] and [35] are Conference Proceedings, they were referenced according to the information available, following the instructions of "Reference List and Citations Style Guide for MDPI Journals".

Round 2

Reviewer 1 Report

1. Page 7, Line 270: According to the results on training data, XGBoost rather than random forest should perform the best. Since new methods are added, the authors need to double check whether previous statements still hold on model comparison;

2. It is better to use four or two decimal places consistently when stating the model comparison results (in tables, figures, and statements);

3. Some figures seem to be too large and their sizes need to be adjusted. 

Author Response

Dear Reviewer 1

We are grateful for the valuable comments that have enriched the study. In response to the comments made to the document, we would like to indicate the following:

  1. Page 7, Line 270: According to the results on training data, XGBoost rather than random forest should perform the best. Since new methods are added, the authors need to double check whether previous statements still hold on model comparison;

Response: The text was replaced by: “The best model in the training was XGBoost with r² = 0.9597, NSE = 0.9567 and RMSE = 0.1513 t/ha, followed by RF with a slightly lower performance”

  1. It is better to use four or two decimal places consistently when stating the model comparison results (in tables, figures, and statements);

Response: The number of decimal places was changed to four places in figure 4, table 4 and numbers within the text.

  1. Some figures seem to be too large and their sizes need to be adjusted. 

Response: The size of figures 1, 2, 3 and 4 was modified to fit the width of the text.
